# Incidence and predictors of extrapulmonary tuberculosis among people living with *Human Immunodeficiency Virus* in Addis Ababa, Ethiopia: A retrospective cohort study

**Ayinalem Alemu**[1]*, **Aman Yesuf**[2], **Ewenat Gebrehanna**[2], **Betselot Zerihun**[1], **Melak Getu**[1], **Teshager Worku**[3], **Zebenay Workneh Bitew**[2]

**1** Ethiopian Public Health Institute, Addis Ababa, Ethiopia, **2** St Paul's Hospital Millennium Medical College, Addis Ababa, Ethiopia, **3** Haramaya University, College of Health and Medical Sciences, Harar, Ethiopia

* ayinalemal@gmail.com

**Data Availability Statement:** All relevant data are within the paper.

## Abstract

### Background

Extrapulmonary tuberculosis is an emerging public health problem among *HIV* positives compared to the general population. This study aimed to assess the incidence and predictors of extrapulmonary tuberculosis among people living with *HIV* in selected health facilities in Addis Ababa, Ethiopia, from 01 January 2013 up to 31 December 2018.

### Methods

A retrospective cohort study design was employed based on data collected from 566 *HIV* positive individuals. Data were entered using EpiInfo version 7.1 and analyzed by SPSS version 20. The incidence rate was determined per 100 person-years. Kaplan-Meier estimates used to estimate survivor and the hazard function, whereas log-rank tests used to compare survival curves and hazard across different categories. Cox proportional hazard model was used to identify the predictors and 95%CI of the hazard ratio were computed. *P-value*<0.05 in the multivariable analysis was considered statistically significant.

### Results

Five hundred sixty-six *HIV* positive individuals were followed for 2140.08 person-years. Among them, 72 developed extrapulmonary tuberculosis that gives an incidence rate of 3.36/100 person-years (95%CI = 2.68–4.22). The most frequent forms of extrapulmonary tuberculosis were; lymph node tuberculosis (56%, 41) followed equally by pleural tuberculosis (15%, 11) and disseminated tuberculosis (15%, 11). The majority (70.83%) of the cases occurred within the first year of follow-up. In multivariable Cox regression analysis, baseline WHO stage III/IV (AHR = 2.720, 95%CI = 1.575–4.697), baseline CD4 count<50cells/µl (AHR = 4.073, 95%CI = 2.064–8.040), baseline CD4 count 50–200 cells/µl (AHR = 2.360, 95%CI = 1.314–4.239) and baseline Hgb<10 mg/dl (AHR = 1.979, 95%CI = 1.091–3.591) were the independent risk factors. While isoniazid prophylaxis (AHR = 0.232, 95%CI =

**Funding:** The author(s) received no specific funding for this work.

**Competing interests:** The authors have declared that no competing interests exist.

**Abbreviations:** AIDS, Acquired Immunodeficiency Syndrome; ART, Anti-Retroviral Therapy; BMI, Body Mass Index; CD4, Cluster of Differentiation; CI, Confidence Interval; CPT, Co-trimoxazole Preventive Therapy; EPTB, Extra-pulmonary Tuberculosis; HAART, Highly Active Anti-Retroviral Treatment; Hgb, Hemoglobin; HIV, Human Immunodeficiency Virus; IPT, Isoniazid Preventive Therapy; MDR, Multi-Drug Resistance; MTB, Mycobacterium Tuberculosis; NEE, North East Ethiopia; NEW, North West Ethiopia; PLHIV, People Living with Human Immunodeficiency Virus; PY, Persons Year; PTB, Pulmonary Tuberculosis; SPSS, Statistical Software for Social Sciences; TB, Tuberculosis; WHO, World Health Organization.

0.095–0.565) and taking antiretroviral drugs (AHR = 0.134, 95%CI = 0.075–0.238) had a protective benefit.

## Conclusion

Extrapulmonary tuberculosis co-infection was common among *HIV* positive individuals, and mostly occurred in those with advanced immune suppression. The risk decreases in those taking antiretroviral therapy and took isoniazid preventive treatment. Screening of *HIV* positives for extrapulmonary tuberculosis throughout their follow-up would be important.

## Introduction

Tuberculosis (TB) is among the top ten causes of death and the leading cause of a single infectious agent worldwide [1]. Even though TB is a global problem, it mainly affects sub-Saharan African countries where the burden of *Human Immunodeficiency Virus (HIV)* is high [2]. There is a strong synergy between TB and *HIV* infection in resource-limited settings [1]. TB is the most frequently diagnosed opportunistic infections[3] and the leading killer among people living with *HIV* (PLHIV) [2]. The main reason for the resurgence of TB in Africa is the link between TB and *HIV* in addition to the lack of adequate economic and human resources [2]. TB facilitates the rapid progression of *HIV* disease [4–6]. In turn, *HIV* increases the lifetime risk of developing TB [7, 8]. In 2017, TB caused an estimated 300,000 deaths among *HIV* positive people. Ethiopia is among high TB, MDR/TB and TB/*HIV* burden countries with an estimated incidence rate of 164 TB cases per 100,000 population [1].

*Mycobacterium tuberculosis (MTB)* mainly affects the lung (pulmonary TB) but it can also affect other parts of the body (extrapulmonary TB) [1, 9]. The majority of TB cases estimated to be pulmonary TB, while in *HIV* positive individuals extrapulmonary tuberculosis (EPTB) has a significant proportion [1, 10]. Extrapulmonary tuberculosis is the isolated occurrence of TB in any part of the body other than the lungs. *MTB* may spread to any organ of the body through lymphatic or hematogenous dissemination and lay dormant for years at a particular site before causing disease [11, 12].

Even though EPTB is mostly common in *HIV* positives, it does not receive specific attention since it does not contribute significantly to the transmission of the disease. That is why national tuberculosis control programs are highly focused on pulmonary TB. Likewise, the majority of the previous studies done in Ethiopia mainly focused on the pulmonary type of TB. The information on the incidence of EPTB and its predictors among PLHIV is limited in the current setting. However, it is important to prevent and control EPTB related morbidity and mortality among *HIV* positive individuals. This study aimed to assess the incidence and predictors of EPTB among people living with *HIV* in selected public health facilities in Addis Ababa registered newly from 01 January 2013 up to 31 December 2013.

## Methods

### Study setting

Retrospective data were collected from the patient's chart and registration books of seven antiretroviral therapy (ART) clinics found in Addis Ababa, Ethiopia. Addis Ababa is the capital city of Ethiopia and the most populous city in the country. In the city, there are more than 123

ART clinics that provide treatment, care, and support to PLHIV. Among them, 18 ART centers in public health facilities had high patient flow.

## Study design and period

An institution-based retrospective cohort study design was conducted. Data were collected in a period between July and August 2019.

## Participants

All people living with *HIV* free of EPTB and newly registered to ART clinics from 01 January 2013 to 31 December 2013 were included in the study and were followed retrospectively for five years up 31 December 2018. Individuals who didn't have baseline data and missed charts were excluded.

## Sample size and sampling procedures

The sample size was calculated using double population proportion formula by using EpiInfo version 7.1 considering 95% confidence level, 80% power, 21.5% proportion among *HIV* positive patients with CD4 count<200 cells/μl and 11% proportion among *HIV* positive patients with CD4 count>500 cells/μl at time of *HIV* diagnosis [13]. With an assumption of equal sample size for exposed and non-exposed groups, a design effect of 1.5 and 5% for incomplete baseline data and missed charts a total sample size of 612 was determined.

A multi-stage sampling technique was followed. Eighteen health facilities that were identified to have a high patient flow were selected in the first stage purposely to get enough proportional samples by assuming a homogenous study population. Among these 18 ART clinics, seven were randomly selected using a simple random sampling technique. The selected ART clinics were; St. Paul's Hospital Millennium Medical College (SPHMMC), St. Peter Hospital, *Ras Desta Damtew* hospital, *Arada* Health Center, *Kirkose* Health Center, *Kolfe* Health Center, and *Addis Ketema* Health Center. A total of 612 patient records included in the data collection using a systematic random sampling method based on the proportion of registered patients in each facility.

## Variables

The outcome of the study was time to event of extrapulmonary tuberculosis in 100 person-years. The independent variables were socio-demographic characteristics, behavioral factors, and clinical factors. Socio-demographic characteristics include; age, sex, marital status, educational status, occupation, address, disclosure status, number of family members and homeless. Behavioral factors include; addiction to smoking, excessive alcohol use, *Khat* and substance use (*Shisha*). Clinical factors were; TB treatment history, functional status, WHO clinical stage, CD4 count, hemoglobin level, enrolled on ART, initial ART regimen type, isoniazid preventive treatment (IPT), co-trimoxazole preventive treatment (CPT), co-infection other than TB, body mass index, ART treatment adherence and ART regimen change.

## Data collection and quality control

A structured data extraction form was used for data collection. The form was pre-tested on 5% of the total sample that was not included in the study. Nine data collectors collected the data from the patient's chart and registration books under the control of the principal investigator. Before data collection, training was given to data collectors to integrate the objective of the study, the method used and the contents of the data extraction form.

## Operational definitions

Extrapulmonary tuberculosis operationalized as any bacteriological, histological or clinical diagnosis of EPTB during the follow-up period. PLHIV who developed EPTB during the follow-up period were considered as events, while those who transferred out, dead, dropped or alive on follow-up who did not develop EPTB were considered as censored. The time to develop EPTB was considered as EPTB free survival time.

Disclosure status: if there is anyone else who knows the *HIV* status of the patient it is operationalized as disclosed.

Any level of alcohol use and smoking was operationalized as a user.

Functional status is the condition of the patient at the time of enrollment in ART clinic categorized as to whether working (able to perform usual work), ambulatory (able to perform the activity of daily living) or bedridden functional status (not able to perform the activity of daily living).

IPT completion: taking the complete prophylaxis, such that a dose of 300mg/day isoniazid for six months.

ART interruption: any interruption of taking ART whether a single day or more was considered an interruption.

ART adherence: it is categorized as good when the patient misses three or fewer doses, fair when the patient misses between three and eight doses and poor when the patient missing more than eight doses per month.

## Data processing and analysis

Data were entered using EpiInfo version 7.1 Software and exported to SPSS version 20.0 Software for analysis. Descriptive summary statistics used to characterize different variables. The event of interest was EPTB incidence. The incidence rate determined per 100 person-years (PYs). The EPTB incidence rate per 100 PYs of follow-up in each category was determined by dividing the number of EPTB in each category to the total person-years of the category and then multiplying by 100. The 95%CI for the incidence rate is determined using the formula $95\%CI = e^{\ln \text{ incidence rate} - Z\alpha/_2 * SE}, e^{\ln \text{ incidence rate} + Z\alpha/_2 * SE}$. Kaplan-Meier estimates and log-rank tests were used to describe time-to-event distributions and to compare time-to-event across the different categories respectively. Bi-variable and multivariable Cox proportional hazard models computed to identify the predictors of EPTB incidence. Variables with *P-value* < 0.25 in the bi-variable analysis entered into the multivariable Cox-proportional model. A 95% CI of the hazard ratio computed, and those variables with *P-value* < 0.05 in the multivariable analysis were considered statistically significant predictors of EPTB. Schoenfield residual test was used to assess Cox proportional hazard assumptions.

## Ethical consideration and consent

Ethical clearance was obtained from St. Paul's Hospital Millennium Medical College Ethical Review Committee, Addis Ababa City Administration Health Bureau and Saint Peter Specialized Hospital. Permission was obtained from all participating health facilities. Study unique identifier number was used in the entire data collection process, not to use patient identifiers. Since it is a retrospective study, obtaining consent to participate was not available.

## Results

### Socio-demographic and behavioral characteristics

Data were extracted from a total of 612 *HIV* positive patients' charts. Among all, 46 excluded from the analysis due to not having a follow-up time. Therefore, the data extracted from 566

patient charts included in the analysis (**Fig 1**). Majorities were females (385, 68%) and the mean age was 35.44 (±8.93) years. Married ones (260, 45.9%) contributed the largest number. More than half of the participants (305, 53.9%) were either not have formal education or completed primary school. Similarly, more than half (330, 58.5%) did not have work. Most of the participants resided in Addis Ababa (529, 93.5%). More than half (358, 63.3%) had an addiction at least to one substance (tobacco or alcohol or *khat* or *shisha*) (**Table 1**).

## Clinical characteristics of study participants

Around nine percent (53, 9.4%) of study participants had an experience of TB treatment history before they knew their *HIV* serostatus. Most (488, 86.2%) were found in a working baseline functional status during enrollment. Similarly, majorities categorized under WHO stage I or II (513, 91.2%) at baseline. The baseline CD4 count of 45.6% (258) of the study participants was <200 cells/μl with a median count of 218 (IQR = 116–321) cells/μl. One third (182) had a BMI value of <18.5 kg/m$^2$ at the baseline with a mean value of 20.74 kg/m$^2$ (±3.76) and the mean baseline Hgb value was 13.32 g/dl (±2.21). Majorities (84.1%, 476) were enrolled on HAART during the follow-up period. Less than half took a complete IPT (44.5%, 252), but majorities took CPT (84.1%, 476). Nearly half (46.8%, 265) were co-infected with different infections other than TB (**Table 2**).

## Incidence of extrapulmonary tuberculosis

A total of 566 *HIV* positive patients were followed for 2140.08 PYs (95%CI = 2.68–4.22). Among all, 72 (12.7%) developed EPTB with an incidence density of 3.36 per 100PYs. Among the remaining 494 patients who did not develop EPTB, 420 were on follow-up, 48 were lost to follow-up, 21 were transferred out and 5 died. The most frequent forms of EPTB were lymph nodes TB (56%, 41) followed equally by pleural TB (15%, 11) and disseminated TB (15%, 11). Other forms were abdominal TB, bone and joint TB, TB of the central nervous system and pericardial TB (**Fig 2**). Bacteriological, histological and clinical methods were used for the diagnosis

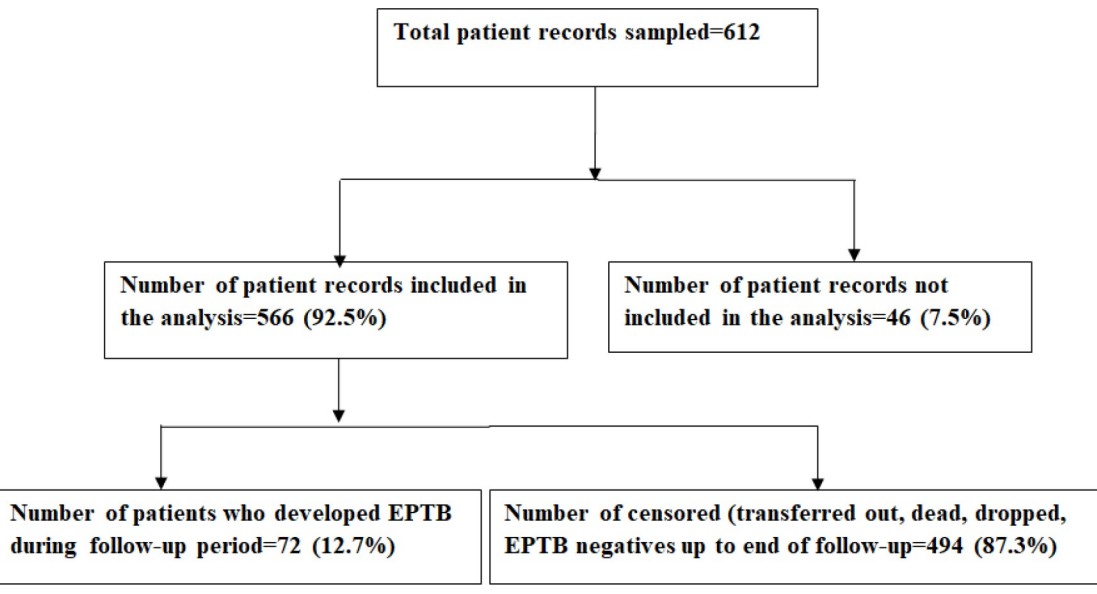

**Fig 1. Flow chart, followed to assess the incidence of EPTB among *HIV* positive patients in Addis Ababa, Ethiopia, from 01 January 2013 up to 31 December 2018 (n = 612).**

**Table 1. Socio-demographic and behavioral characteristics among *HIV* positive patients in Addis Ababa, Ethiopia, from 01 January 2013 up to 31 December 2018 (n = 566).**

| Characteristics | Number | Proportion |
|---|---|---|
| **Age group** | | |
| 15–24 | 38 | 6.7 |
| 25–34 | 230 | 40.6 |
| 35–44 | 205 | 36.2 |
| >44 | 93 | 16.4 |
| **Sex** | | |
| Female | 385 | 68 |
| Male | 181 | 32 |
| **Marital status** | | |
| Single | 127 | 22.4 |
| Married | 260 | 45.9 |
| Separated/Divorced | 114 | 20.1 |
| Widowed | 65 | 11.5 |
| **Educational status** | | |
| No formal education | 100 | 17.7 |
| Primary | 206 | 36.4 |
| Secondary | 195 | 34.5 |
| Tertiary | 65 | 11.5 |
| **Employment status** | | |
| Yes | 236 | 41.7 |
| No | 330 | 58.3 |
| **Address** | | |
| Addis Ababa | 529 | 93.5 |
| Out of Addis Ababa | 37 | 6.5 |
| **Disclosure status** | | |
| Yes | 404 | 71.4 |
| No | 162 | 28.6 |
| **Family size** | | |
| 1–3 | 334 | 59.0 |
| 4–5 | 165 | 29.2 |
| >5 | 67 | 11.8 |
| **Homeless** | | |
| Yes | 4 | 0.7 |
| No | 562 | 99.3 |
| **Tobacco smoking** | | |
| Yes | 278 | 49.1 |
| No | 288 | 50.9 |
| **Alcohol addiction** | | |
| Yes | 312 | 55.1 |
| No | 254 | 44.9 |
| **Taking Khat** | | |
| Yes | 296 | 52.3 |
| No | 270 | 47.7 |
| **Taking Hard drugs/Shisha** | | |
| Yes | 270 | 47.7 |
| No | 296 | 52.3 |

**Table 2. Baseline and follow-up clinical characteristics among *HIV* positive patients in Addis Ababa, Ethiopia, from 01 January 2013 up to 31 December 2018 (n = 566).**

| Characteristics | Number | Proportion |
|---|---|---|
| Previous TB history | | |
| Yes | 53 | 9.4 |
| No | 513 | 90.6 |
| Baseline functional status | | |
| Working | 488 | 86.2 |
| Ambulatory | 70 | 12.4 |
| Bedridden | 8 | 1.4 |
| Baseline WHO stage | | |
| I /II | 513 | 91.2 |
| III /IV | 53 | 9.4 |
| Baseline CD4 count | | |
| <50 | 47 | 8.3 |
| 50–200 | 211 | 37.3 |
| >200 | 308 | 54.4 |
| Baseline Hgb | | |
| <10g/dl | 51 | 9.0 |
| >10g/dl | 515 | 91.0 |
| BMI | | |
| <18.5 | 182 | 32.2 |
| >18.5 | 384 | 67.8 |
| On HAART | | |
| Yes | 476 | 84.1 |
| No | 90 | 15.9 |
| Initial treatment | | |
| TDF/3TC/NVP | 14 | 2.94 |
| AZT/3TC/NVP | 56 | 11.76 |
| AZT/3TC/EFV | 39 | 8.19 |
| TDF/3TC/EFV | 367 | 77.10 |
| Took IPT Prophylaxis | | |
| Yes | 252 | 44.5 |
| No | 314 | 55.5 |
| Took Co-trimoxazole | | |
| Yes | 476 | 84.1 |
| No | 90 | 15.9 |
| Co-infection | | |
| Yes | 265 | 46.8 |
| No | 301 | 53.2 |
| ART Treatment Interruption | | |
| Yes | 14 | 2.9 |
| No | 462 | 97.1 |
| ART adherence | | |
| Good | 459 | 96.4 |
| Fair | 5 | 1.1 |
| Poor | 12 | 2.5 |
| ART regimen change | | |
| Yes | 8 | 1.7 |

*(Continued)*

**Table 2.** (Continued)

| Characteristics | Number | Proportion |
|---|---|---|
| No | 468 | 98.3 |

TB; Tuberculosis, WHO; World Health Organization, Hgb; hemoglobin, BMI; Body Mass Index, HAART; Highly Active Anti-Retroviral Treatment, IPT; Isoniazid Preventive Therapy, ART; Anti-Retroviral Therapy, TDF; Tenofovir, 3TC; Lamivudine, NVP; Nevirapine, AZT; Zidovudine, EFV; Efavirenz, co-infection; the presence of is any comorbidity or infection other than tuberculosis.

of EPTB. Accordingly, nine results were reported positive by bacteriological methods (Culture; 3, Xpert MTB/RIF assay; 6). Fifty-five results were confirmed by histological diagnosis and 16 test results were confirmed based on a combination of clinical and chest radiography. However, different combinations of the above confirmatory methods were used to rule out EPTB.

Most of EPTB cases (51, 70.83%) occurred within the first year of follow-up. The incidence density was 188/100 PYs, 14.29/100PYs, 11.63/100 PYs, 5.56/100 PYs and 0.41/100 PYs at the end of first, second, third, fourth and fifth years of follow-up respectively (**Table 3**). The cumulative probability of EPTB-free survival at the end of, first, two, three, four and five years were 0.91, 0.90, 0.89, 0.88 and 0.87 respectively (**Fig 3**). In terms of survival curves, there were significant variations among baseline WHO stage III or IV and WHO I or II *(P<0.001)*, different baseline CD4 counts *(P<0.001)*, baseline Hgb value <10mg/dl and >10mg/dl *(P<0.001)* and enrolled and not enrolled on HAART *(P<0.001)* (**Fig 4**).

Women contributed the highest proportion (42, 58.33%) however the incidence density was higher in males (Males; 4.76/100PY, Women; 2.78/100PY). Among all 72 EPTB cases, 10

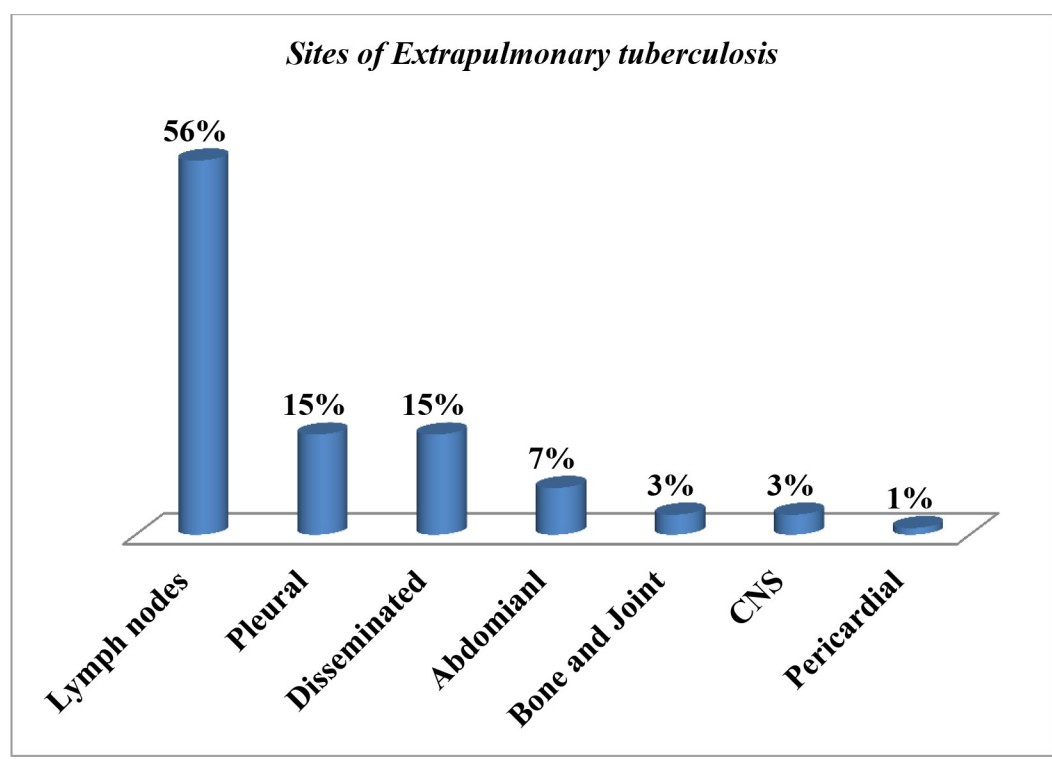

**Fig 2. Sites of extrapulmonary TB among *HIV* positive patients in Addis Ababa, Ethiopia, from 01 January 2013 up to 31 December 2018 (n = 72).**

**Table 3. Extrapulmonary tuberculosis incidence density among *HIV* positive patients in Addis Ababa, Ethiopia, from 01 January 2013 up to 31 December 2018 (n = 566).**

| Characteristics | Total | EPTTB | Percent | Person years | EPTB incidence rate(95%CI) |
|---|---|---|---|---|---|
| **Age group** | | | | | |
| 15–24 | 38 | 3 | 4.17 | 125.13 | 2.40(0.78–7.33) |
| 25–34 | 230 | 22 | 30.56 | 913.78 | 2.41(1.59–3.64) |
| 35–44 | 205 | 34 | 47.22 | 745.53 | 4.56(3.28–6.33) |
| >44 | 93 | 13 | 18.06 | 355.64 | 3.66(2.14–6.23) |
| **Sex** | | | | | |
| Female | 385 | 42 | 58.33 | 1510.16 | 2.78(2.06–3.75) |
| Male | 181 | 30 | 41.67 | 629.92 | 4.76(3.36–6.75) |
| **Marital status** | | | | | |
| Single | 127 | 16 | 22.22 | 483.27 | 3.31(2.04–5.36) |
| Married | 260 | 33 | 45.83 | 1012.39 | 3.26(2.33–4.56) |
| Separated/Divorced | 114 | 17 | 23.61 | 376.68 | 4.51(2.84–7.18) |
| Widowed | 65 | 6 | 8.33 | 267.74 | 2.24(1.02–4.94) |
| **Educational status** | | | | | |
| No formal education | 100 | 13 | 18.06 | 393 | 3.31(1.94–5.65) |
| Primary | 206 | 24 | 33.33 | 772.09 | 3.117(2.10–4.61) |
| Secondary | 195 | 24 | 33.33 | 728.32 | 3.30(2.22–4.88) |
| Tertiary | 65 | 11 | 15.28 | 246.32 | 4.47(2.51–7.96) |
| **Employment status** | | | | | |
| Yes | 236 | 26 | 36.11 | 929.43 | 2.80(1.92–4.09) |
| No | 330 | 46 | 63.89 | 1210.65 | 3.80(2.86–5.04) |
| **Address** | | | | | |
| Addis Ababa | 529 | 65 | 90.28 | 1987.25 | 3.27(2.58–4.15) |
| Out of Addis Ababa | 37 | 7 | 9.72 | 152.82 | 4.58(2.22–9.44) |
| **Disclosure status** | | | | | |
| Yes | 404 | 52 | 72.22 | 1529.30 | 3.40(2.60–4.44) |
| No | 162 | 20 | 27.78 | 610.78 | 3.27(2.13–5.04) |
| **Family size** | | | | | |
| 1–3 | 334 | 43 | 59.72 | 1271 | 3.38(2.52–4.54) |
| 4–5 | 165 | 20 | 27.78 | 643 | 3.11(2.02–4.79) |
| >5 | 67 | 9 | 12.50 | 226 | 3.98(2.10–7.55) |
| **Homeless** | | | | | |
| Yes | 4 | 1 | 1.39 | 11.40 | 8.77(1.35–57.03) |
| No | 562 | 71 | 98.61 | 2128.68 | 3.34(2.65–4.19) |
| **Tobacco smoking** | | | | | |
| Yes | 278 | 32 | 44.44 | 1028 | 3.11(2.21–4.38) |
| No | 288 | 40 | 55.56 | 1112 | 3.60(2.65–4.88) |
| **Alcohol** | | | | | |
| Yes | 312 | 39 | 54.17 | 1161 | 3.36(2.47–4.57) |
| No | 254 | 33 | 45.83 | 979 | 3.37(2.41–4.71) |
| **Taking Khat** | | | | | |
| Yes | 296 | 34 | 47.22 | 1114 | 3.05(2.19–4.25) |
| No | 270 | 38 | 52.78 | 1026 | 3.70(2.71–5.06) |
| **Taking hard drugs/Shisha** | | | | | |
| Yes | 270 | 30 | 41.67 | 1015 | 2.96(2.08–4.20) |
| No | 296 | 42 | 58.33 | 1125 | 3.73(2.77–5.02) |

*(Continued)*

**Table 3.** (Continued)

| Characteristics | Total | EPTTB | Percent | Person years | EPTB incidence rate(95%CI) |
|---|---|---|---|---|---|
| **Previous TB history** | | | | | |
| Yes | 53 | 10 | 13.89 | 175.70 | 5.69(3.12–10.39) |
| No | 513 | 62 | 86.12 | 1964.37 | 3.16(2.47–4.03) |
| **Baseline functional status** | | | | | |
| Working | 488 | 51 | 70.83 | 1966.74 | 2.59(1.98–3.40) |
| Ambulatory | 70 | 19 | 26.39 | 162.73 | 11.66(7.65–17.82) |
| Bedridden | 8 | 2 | 2.78 | 10.60 | 18.87(5.41–65.75) |
| **Baseline WHO stage** | | | | | |
| I/II | 513 | 46 | 63.89 | 2060.16 | 2.23(1.68–2.97) |
| III/ IV | 53 | 26 | 36.11 | 79.92 | 32.53(23.47–44.61) |
| **Baseline CD4 count** | | | | | |
| <50 | 47 | 20 | 27.78 | 96.03 | 20.83(14.10–30.76) |
| 50–200 | 211 | 34 | 47.22 | 723.97 | 4.70(3.38–6.52) |
| >200 | 308 | 18 | 25.0 | 1320.08 | 1.36(0.86–2.16) |
| **Baseline Hgb** | | | | | |
| <10g/dl | 51 | 18 | 25.0 | 94 | 19.15(12.64–29.01) |
| >10g/dl | 515 | 54 | 75.0 | 2046 | 2.64(2.03–3.43) |
| **BMI** | | | | | |
| <18.5 | 182 | 33 | 45.83 | 532 | 6.20(4.46–8.63) |
| >18.5 | 384 | 39 | 54.17 | 1608 | 2.43(1.78–3.31) |
| **On HAART** | | | | | |
| Yes | 476 | 32 | 44.44 | 2056.93 | 1.56(1.10–2.19) |
| No | 90 | 40 | 55.56 | 83.15 | 48.11(38.48–60.14) |
| **Initial treatment** | | | | | |
| TDF/3TC/NVP | 14 | 2 | 6.25 | 54 | 3.70(0.95–14.43) |
| AZT/3TC/NVP | 56 | 4 | 12.50 | 241 | 1.66(0.63–4.39) |
| AZT/3TC/EFV | 39 | 0 | 0.00 | 185 | - |
| TDF/3TC/EFV | 367 | 26 | 81.25 | 1582 | 1.64(1.12–2.41) |
| **Took IPT Prophylaxis** | | | | | |
| Yes | 252 | 6 | 8.33 | 1206.81 | 0.50(0.22–1.10) |
| No | 314 | 66 | 91.67 | 933.27 | 7.07(5.60–8.92) |
| **Took Co-trimoxazole** | | | | | |
| Yes | 476 | 45 | 62.50 | 1988.59 | 2.26(1.70–3.02) |
| No | 90 | 27 | 37.50 | 151.48 | 17.82(12.66–25.06) |
| **Co-infection** | | | | | |
| Yes | 265 | 36 | 50.0 | 964.35 | 3.73(2.71–4.22) |
| No | 301 | 36 | 50.0 | 1175.73 | 3.06(2.22–4.22) |
| **ART Treatment Interruption** | | | | | |
| Yes | 14 | 1 | 3.13 | 53.44 | 1.87(0.27–13.04) |
| No | 462 | 31 | 96.87 | 2008.49 | 1.54(1.09–2.19) |
| **ART adherence** | | | | | |
| Good | 459 | 30 | 93.75 | 2007.83 | 1.49(1.05–2.13) |
| Fair | 5 | 0 | 0 | 18.18 | - |
| Poor | 12 | 2 | 6.25 | 35.92 | 5.57(1.45–21.41) |
| **ART regimen change** | | | | | |
| Yes | 8 | 1 | 3.13 | 35.67 | 2.8(0.41–19.36) |
| No | 468 | 31 | 96.87 | 2026.27 | 1.53(1.08–2.17) |

*(Continued)*

**Table 3.** (Continued)

| Characteristics | Total | EPTTB | Percent | Person years | EPTB incidence rate(95%CI) |
|---|---|---|---|---|---|
| **Follow-up year** | | | | | |
| 1 | 114 | 51 | 70.83 | 27 | - |
| 2 | 19 | 4 | 5.56 | 28 | 14.29(5.77–35.39) |
| 3 | 16 | 5 | 6.94 | 43 | 11.63(5.10–26.51) |
| 4 | 21 | 4 | 5.56 | 72 | 5.56(2.14–14.40) |
| 5 | 396 | 88 | 11.1 | 1970 | 0.41(3.64–5.48) |

EPTB; Extrapulmonary tuberculosis, TB; Tuberculosis, WHO; World Health Organization, Hgb; hemoglobin, BMI; Body Mass Index, HAART; Highly Active Anti-Retroviral Treatment, IPT; Isoniazid Preventive Therapy, ART; Anti-Retroviral Therapy, TDF; Tenofovir, 3TC; Lamivudine, NVP; Nevirapine, AZT; Zidovudine, EFV; Efavirenz, co-infection; the presence of is any comorbidity or infection other than tuberculosis.

(13.89%) had a previous TB treatment history, 21 (29.17%) were either ambulatory or bedridden functional status at enrollment, 26 (36.11%) were either at WHO stage III or IV, 54 (75%) had CD4 count <200 cells/μl, 18 (25%) had Hgb value <10 mg/dl, 33 (45.83%) were underweight, 40 (55.56%) did not take ART drugs, 66 (91.67%) did not receive IPT and 27 (37.5%) did not take the CPT (**Table 3**).

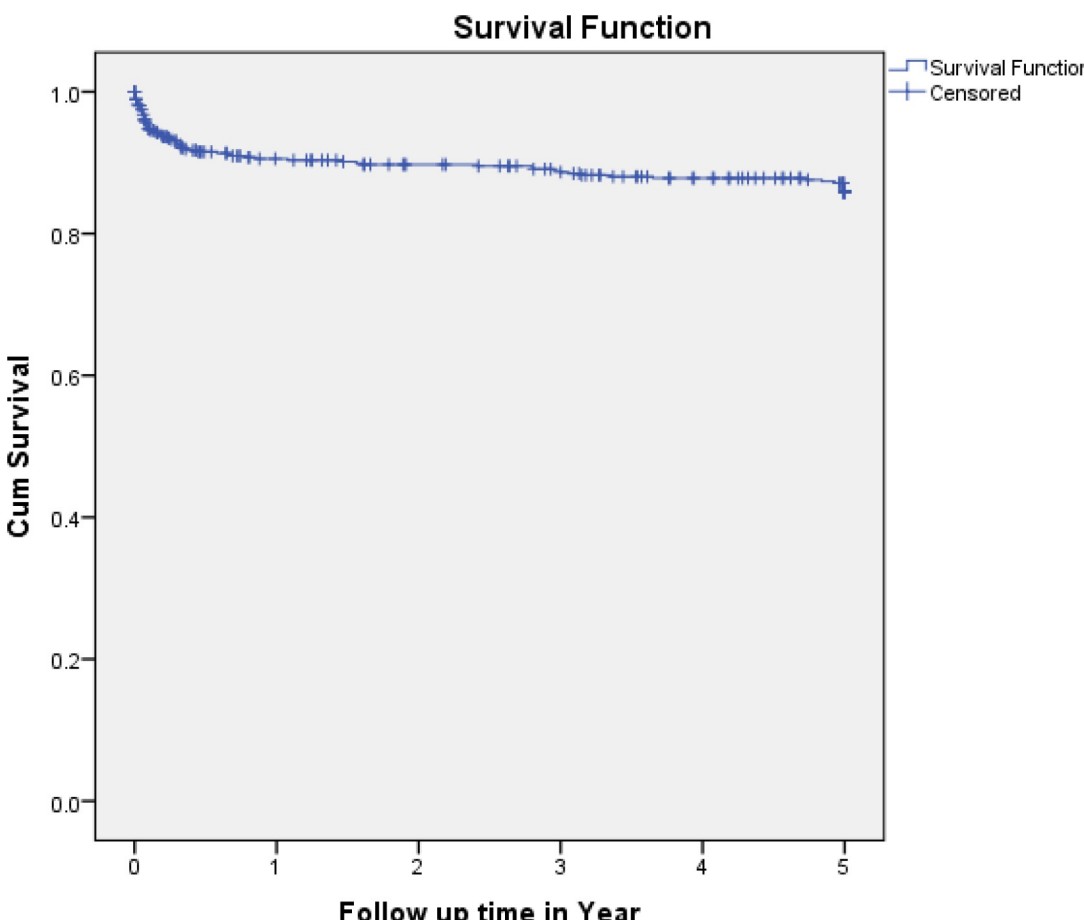

**Fig 3. Kaplan-Meier EPTB free survival curves among *HIV* positive patients in Addis Ababa, Ethiopia, from 01 January 2013 up to 31 December 2018 (n = 566).**

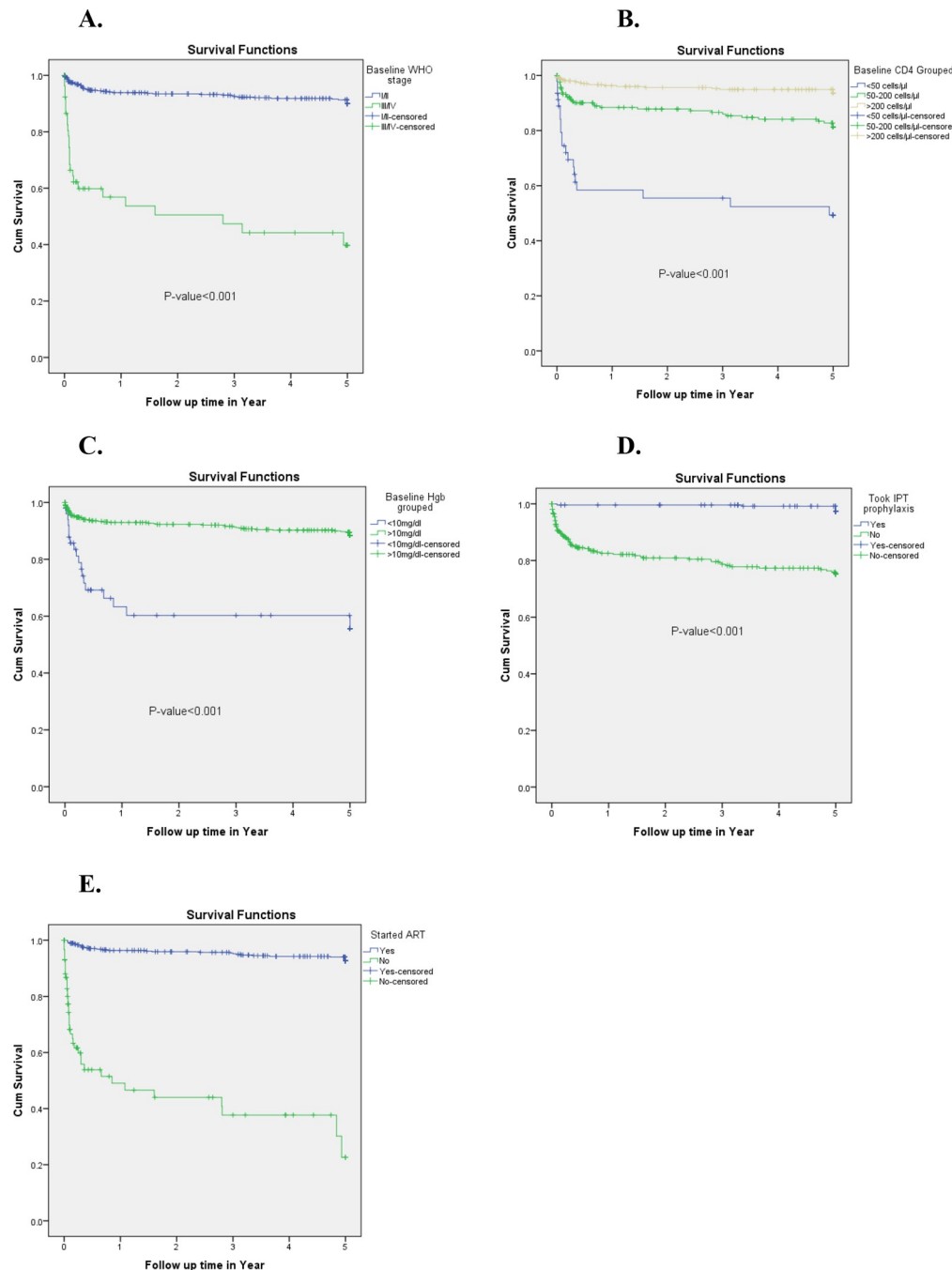

**Fig 4.** Kaplan-Meier survival curves of patients with extrapulmonary tuberculosis based on baseline WHO stage (A), baseline CD4 (B) count, baseline anemia (C), IPT intake(D) and enrollment on HAART (E) among *HIV* positive patients in Addis Ababa, Ethiopia, from 01 January 2013 up to 31 December 2018 (n = 566).

## Predictors of extrapulmonary tuberculosis

In the bi-variable Cox regression analysis, age-group, sex, previous anti-TB treatment, baseline functional status, baseline WHO stage, baseline CD4 count, baseline Hgb value, baseline BMI, HAART status, IPT, and CPT were found as risk factors of the EPTB at a *P-value* of less than 0.25. When all these variables were subjected to multivariable Cox regression analysis, WHO

**Table 4. Bi-variable and multi-variable Cox regression analysis of extrapulmonary tuberculosis among *HIV* positive patients in Addis Ababa, Ethiopia, from 01 January 2013 up to 31 December 2018 (n = 566).**

| Characteristics | Survival status | | CHR(95% CI) | AHR(95%CI) |
|---|---|---|---|---|
| | Event | Censored | | |
| Age group | | | | |
| 15–24 | 3 | 35 | 0.598(0.170–2.098) | 0.742(0.203–2.704) |
| 25–34 | 22 | 208 | 0.660(0.332–1.309) | 1.232(0.576–2.638) |
| 35–44 | 34 | 171 | 1.1212(0.640–2.296) | 1.510(0.771–2.958) |
| >44 | 13 | 80 | 1 | 1 |
| Sex | | | | |
| Female | 42 | 343 | 1 | 1 |
| Male | 30 | 151 | 1.644(1.029–2.627) | 0.928(0.555–1.553) |
| Previous TB history | | | | |
| Yes | 10 | 43 | 1.619(0.830–3.160) | 1.437(0.731–2.825) |
| No | 62 | 451 | 1 | 1 |
| Baseline functional status | | | | |
| Working | 51 | 437 | 1 | 1 |
| Ambulatory | 19 | 51 | 3.684 (2.170–6.254) | 0.875(0.480–1.597) |
| Bedridden | 2 | 6 | 5.012(1.216–20.668) | 0.534(0.108–2.652) |
| Baseline WHO stage | | | | |
| I and II | 46 | 467 | 1 | 1 |
| III and IV | 26 | 27 | 9.997(6.103–16.310) | 2.720(1.575–4.697) |
| Baseline CD4 count | | | | |
| <50 | 20 | 27 | 11.850(6.248–22.474) | 4.073(2.064–8.040) |
| 50–200 | 34 | 177 | 3.514(1.781–5.588) | 2.360(1.314–4.239) |
| >200 | 18 | 290 | 1 | 1 |
| Baseline Hgb | | | | |
| <10g/dl | 18 | 33 | 4.869(2.839–8.350) | 1.979(1.091–3.591) |
| >10g/dl | 54 | 461 | 1 | 1 |
| Baseline BMI | | | | |
| <18.5 | 33 | 149 | 2.227(1.399–3.544) | 1.146(0.680–1.933) |
| >18.5 | 39 | 345 | 1 | 1 |
| On HAART | | | | |
| Yes | 32 | 444 | 1.00 | 1 |
| No | 40 | 50 | 19.865 (12.028–32.807) | 7.645(4.201–13.263) |
| Taken IPT Prophylaxis | | | | |
| Yes | 6 | 246 | 1 | 1 |
| No | 66 | 248 | 11.897(5.151–27.479) | 4.314(1.769–10.520) |
| Taken Co-trimoxazole | | | | |
| Yes | 45 | 431 | 1 | 1 |
| No | 27 | 63 | 5.926(3.657–9.603) | 1.484(0.783–2.814) |

CHR; crude hazard ratio, AHR; adjusted hazard ratio, TB; Tuberculosis, WHO; World Health Organization, Hgb; hemoglobin, BMI; Body Mass Index, HAART; Highly Active Anti-Retroviral Treatment, IPT; Isoniazid Preventive Therapy, ART; Anti-Retroviral Therapy, co-infection; the presence of is any comorbidity or infection other than tuberculosis.

stage III or IV, baseline CD4 count<50 cells/μl, baseline CD4 count 50–200 cells/μl, baseline Hgb<10 mg/dl, not on HAART, and not received IPT were found to be the statistically significant independent risk factors (**Table 4**).

Accordingly, PLHIV who were on WHO stage III or IV at the time of enrollment had 2.72 times the risk to develop EPTB at any time compared to those who were on WHO stage I or II (AHR = 2.720, 95%CI = 1.575–4.697). PLHIV who had a baseline CD4 count of <50 cells/μl and a CD4 count of 50–200 cells/μl were 4.073 times (AHR = 4.073, 95% CI = 2.064–8.040) and 2.36 times (AHR = 2.360, 95%CI = 1.314–4.239) as likely to have EPTB compared to PLHIV who had a baseline CD4 count of >200 cells/μl respectively. Similarly, PLHIV who had a baseline Hgb value <10 mg/dl were 1.98 times the risk to develop EPTB at any time compared to PLHIV who had a baseline Hgb value>10 mg/dl (AHR = 1.979, 95%CI = 1.091–3.591). Also, the risk of developing EPTB among those who did not receive IPT was 4.314 times as high as the risk to develop EPTB among those who took IPT (AHR = 4.314, 95% CI = 1.769–10.520). While, those who took IPT had a 77% reduced risk of EPTB (AHR = 0.232, 95%CI = 0.095–0.565). Similarly, PLHIV who did not enroll on HAART were 7.645 times as likely to develop EPTB compared to those who were on HAART (AHR = 7.645, 95%CI = 4.201–13.263). Such that taking ART drugs reduced the risk of EPTB by 87% (AHR = 0.134, 95%CI = 0.075–0.238) (Table 4).

## Discussion

In the current study, about 12.7% (incidence rate; 3.36/100PYs) of *HIV* positive individuals developed EPTB in their follow-up. Advanced immune suppression at baseline such as an advanced WHO stage, lower CD4 count and anemia were the independent risk factors. However, taking antiretroviral therapy and IPT had a protective benefit.

In the present study, a higher incidence density rate of EPTB was observed among PLHIV. This finding was higher compared to a study done in Northeast Ethiopia [7], in Gondar [2] and Addis Ababa [3]. However, when considering the proportion of EPTB among all *HIV* patients, ours' finding (12.72%) was lower compared to a study done in Addis Ababa [3]. In this study, the highest incidence of EPTB was observed in the first year of follow up and decreased in the consecutive years, which were supported by previous studies [2] [3, 7, 14]. The possible explanation could be the late *HIV* diagnosis and might be due to immune reconstitution inflammatory response (IRIS) which is a common phenomenon in *HIV* positive individuals. After starting HAART the level of CD4 count increased that causes pathogen-specific immune responses/inflammation/ which results in enlargement of lymph nodes. As was reported by previous studies done in different countries and settings, [10, 15] [16–19], the most frequent forms of EPTB in this study were lymph nodes TB. However, in a study done in Nigeria, the commonest forms of EPTB was abdominal TB [20]. The second most frequent forms of EPTB observed in our study were pleural TB and disseminated TB, which were also reported from the current study setting [15] and USA [16, 18] The other forms of EPTB such as abdominal TB, bone and joint TB, TB of the central nervous system and pericardial TB observed in the current study were also reported by previous studies [10, 16–20]. Among 72 EPTB cases, only nine were confirmed by bacteriological methods where the diagnosis of EPTB is difficult due to the paucibacillary nature of the disease.

In the current study, those PLHIV who had a previous TB treatment history had a higher incidence of EPTB compared to those not had TB infection history that was also reported before [7, 21]. The baseline functional status of *HIV* patients during enrollment is a key factor for the likely hood of subsequent progression of different diseases [13, 22]. In our study based on the bi-variable analysis, those PLHIV who were on ambulatory at the time of enrollment had a higher risk to develop EPTB compared with those on working functional status (CHR = 3.684, 95%CI = 2.170–6.254). Likewise, this was also observed in bedridden *HIV* positive patients (CHR = 5.012, 95%CI = 1.216–20.668). This might be because *HIV* positive

patients who were on ambulatory or bedridden functional status at baseline might have low CD4 counts that could make them more susceptible to co-infections like tuberculosis. Previous studies also supported this [2, 3, 7, 13, 22, 23]. We observed that the risk to develop EPTB is higher when they are underweight at the baseline, compared to those with normal weight (CHR = 2.227, 95%CI = 1.399–3.544). Similar findings were reported from other studies [6, 7, 23], where nutritional status is a key factor development of TB.

Based on the multivariable Cox regression analysis, the statistically significant independent predictors of EPTB among PLHIV identified in our study were; WHO stage III or IV, baseline CD4 count <50 cells/μl, baseline CD4 count 50–200 cells/μl, baseline Hgb <10 mg/dl, not on HAART and not received IPT prophylaxis. This study revealed that those PLHIV who were in advanced WHO stages III or IV at the time of enrollment had 2.72 times the risk to develop EPTB at any time compared to those who were on WHO stage I or II. Similarly, different studies reported TB was more common in advanced *HIV* stages [2, 3, 6, 7, 13, 14, 20, 22–29].

Immune status is a key factor for the susceptibility of a host for TB, such that when the immune system becomes week the likely hood to be infected with TB increases [6] [24]. Based on this study and findings from previous studies done in Ethiopia and other countries, individuals who had lower CD4 count at the time of enrollment had higher risk to develop TB [2, 6, 14, 20, 21, 23, 26–33]. The other independent predictor of EPTB observed in this study was the baseline hemoglobin value. It is observed that PLHIV who were anemic (Hgb value <10 mg/dl) had 98% greater risk to develop EPTB compared to PLHIV who were not anemic at the baseline (Hgb value>10 mg/dl). This was also supported by previous studies [6, 7, 22, 25]. Anemia might be indirectly associated with EPTB, where immune compromisation leads to decreased production of red blood cells and EPTB among *HIV* positives.

In the current study, PLHIV who took IPT had a 77% reduced risk of EPTB compared to those who did not take IPT (AHR = 0.232, 95%CI = 0.095–0.565). This finding supported the WHO recommendation; *HIV* patients should receive IPT to decrease the risk of developing TB. This is also recommended in Ethiopia [26]. Likewise, a reduced risk of TB among PLHIV who has taken IPT was reported by different scholars [5, 7, 22, 26, 27, 29, 34–37]. The other independent predictor of EPTB infection observed in this study was not enrolled on HAART. PLHIV who were on HAART had 87% reduced risk of EPTB compared to those who were not enrolled on HAART (AHR = 0.134, 95%CI = 0.075–0.238). In line with this, other studies also reported the significant decrease risk of EPTB among *HIV* positive individuals who were enrolled on HAART [6, 22, 30, 32–35].

## Limitation of the study

This study has important limitations as it is a retrospective study of nature. The incomplete records that we excluded from our analysis might introduce selection bias. Besides, we were unable to obtain additional information not captured to the patient's chart or registration book. Moreover, the lack of information about the outcomes in the lost and transferred out is another limitation.

## Conclusion

Extrapulmonary tuberculosis is the major opportunistic infection among people living with *HIV*. The most frequent forms were lymph nodes TB followed by the equal incidence of pleural TB and disseminated TB. The majority of EPTB cases were occurring in the first years of follow-up. Being on baseline advanced WHO stages, lower baseline CD4 count, and baseline anemia was found to be the statistically significant independent risk factors of EPTB. While taking isoniazid preventive therapy and enrolled on HAART had the protective benefit.

Screening of *HIV* positives for extrapulmonary tuberculosis throughout their follow-up would be important.

## Acknowledgments

We acknowledge administrators and staff of St. Paul's Hospital Millennium Medical College, Addis Ababa City Administration Health Bureau, Saint Peter Specialized Hospital, *Ras Desta Damtew* hospital, *Arada* Health Center, *Kirkose* Health Center, *Kolfe* Health Center and *Addis Ketema* Health Center for their support and commitment.

## Author Contributions

**Data curation:** Ayinalem Alemu, Aman Yesuf.

**Formal analysis:** Ayinalem Alemu, Aman Yesuf, Teshager Worku.

**Investigation:** Betselot Zerihun, Melak Getu, Zebenay Workneh Bitew.

**Methodology:** Ayinalem Alemu, Aman Yesuf, Ewenat Gebrehanna.

**Software:** Ayinalem Alemu, Teshager Worku, Zebenay Workneh Bitew.

**Writing – original draft:** Ayinalem Alemu.

**Writing – review & editing:** Ayinalem Alemu, Aman Yesuf, Ewenat Gebrehanna.

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
