## [Decision Letter · Decision Letter 0]

17 Mar 2020

PONE-D-20-00998

Incidence and Predictors of Extra Pulmonary Tuberculosis among People Living with Human Immunodeficiency Virus in Addis Ababa, Ethiopia: A Retrospective Cohort study

PLOS ONE

Dear Mr Alemu,

Thank you for submitting your manuscript to PLOS ONE. After careful consideration, we feel that it has merit but does not fully meet PLOS ONE’s publication criteria as it currently stands. Therefore, we invite you to submit a revised version of the manuscript that addresses the points raised during the review process.

We would appreciate receiving your revised manuscript by 15th May 2010. To enhance the reproducibility of your results, we recommend that if applicable you deposit your laboratory protocols in protocols.io, where a protocol can be assigned its own identifier (DOI) such that it can be cited independently in the future. For instructions see: http://journals.plos.org/plosone/s/submission-guidelines#loc-laboratory-protocols

We look forward to receiving your revised manuscript.

Kind regards,

Kwasi Torpey, MD PhD MPH

Academic Editor

PLOS ONE

Additional Editor Comments:

1. Kindly thoroughly copyedit the manuscript for error using a native speaker

2. Why is reference 6 in CAPs?

3. Need use to be consistent referencing style. Some references list only 3 while others list 6. Eg Ref 7 and 9

4. Ref 15. Remove typo 15

2. Please amend your list of authors on the manuscript to ensure that each author is linked to an affiliation. Authors’ affiliations should reflect the institution where the work was done (if authors moved subsequently, you can also list the new affiliation stating “current affiliation:….” as necessary).

Reviewers' comments:

Reviewer's Responses to Questions

**Comments to the Author**

1. Is the manuscript technically sound, and do the data support the conclusions?

Reviewer #1: Partly

Reviewer #2: Partly

2. Has the statistical analysis been performed appropriately and rigorously? 

Reviewer #1: Yes

Reviewer #2: Yes

3. Have the authors made all data underlying the findings in their manuscript fully available?

Reviewer #1: Yes

Reviewer #2: Yes

4. Is the manuscript presented in an intelligible fashion and written in standard English?

Reviewer #1: No

Reviewer #2: No

5. Review Comments to the Author

Reviewer #1: I read with interest this article

My main comment is that the language needs to be revised

Other comments.

1) The authors should provide information on the number of lost to follow up and transferred

2) The authors should also provide information on the modality of diagnosis, in how many was clinical and how many supported by lab/other diagnostic. This also deserves to be included in the discussion since extrapulmonary diagnosis is not easy especially in certain settings

3) The authors should provide a limitation section in the discussion, including the lack of information of outcomes in the lost and transferred

4) Conclusions sections. These are not conclusions, but a summary of the results. The authors should provide appropriate conclusion and recommendations

Reviewer #2: Methods:

1. Please clarify the meaning of "disclosure status".

2. How was addiction to smoking and alcohol defined? Can the authors provide information on smoking status (never vs former vs current), pack-years smoked and quantitative estimate of alcohol consumption? Were these measures associated with EPTB?

3. How was functional status defined/classified?

4. Were longitudinal CD4 and HIV RNA levels available during follow-up? Were these associated with the outcome?

5. How many participants had incident PULMONARY TB disease and how was this handled in the analysis? Also, were prevalent pulmonary TB cases excluded?

6. Please provide the statistical method used to calculate incidence rate and its 95% confidence interval.

7. How was "IPT completion" defined?

8. Are data on TB infection status available?

9. How was "ART interruption" and "ART adherence" defined?

10. Is duration of ART prior to enrollment available? Was this associated with incident EPTB?

11. Did participants who died or were lost to follow-up different from the rest? And was death due to EPTB assessed as an outcome? It may be helpful to consider death and loss-to-follow-up as competing-risks for incident EPTB and analyze these data accordingly.

Results:

1. How many EPTB cases were defined by clinical vs microbiological diagnosis?

2. Please report median and inter-quartile range or mean with standard deviation for all continuous exposures.

3. Please provide % in Figure-1.

4. Table-1 and table-2 may be more informative if stratified by the primary outcome.

5. Is "tobacco smoking" in table-1 ever-smoking? Please clarify.

6. Please include foot-notes to all tables and figures clarifying abbreviations.

7. Please clarify what "co-infection"means in Table-2.

8. Please provide 95% confidence intervals for all incidence rate estimates in the text and tables.

9. Was alcohol included in uni- and multi-variable analysis?

Discussion:

1. Can the authors comment on the heterogeneity of HIV populations across the several ART clinics in Ethiopia and comment on the generalizability of their study findings?

2. It is unclear why IRIS would lead to incident EPTB. Can the authors clarify?

3. The conclusion that male sex and not receiving co-trimoxazole prophylaxis are predictors of incident EPTB is incorrect. This may be due to confounding as seen in the uni- and multi-variable regression. Can the authors comment?

4. Can the authors speculate as to why anemia may be associated with incident EPTB?

5. The authors conclude that "early start of HAART" and "early diagnosis of HIV" should be considered for preventing incident EPTB. This is not supported by the data presented. Can the authors comment on the CD4 cell count at ART initiation (as opposed to enrollment) and whether this as associated with EPTB?

General:

1. There are several typos in the manuscript text and figures. Please review and correct.

2. Recognizing that English may not be the authors primary language, please consider having this manuscript proof-read for grammatical errors and syntax.

6. PLOS authors have the option to publish the peer review history of their article (what does this mean?). If published, this will include your full peer review and any attached files.

Reviewer #1: No

Reviewer #2: No

---

## [Author Response · Author response to Decision Letter 0]

28 Mar 2020

Response to Reviewers

Title: Incidence and Predictors of Extrapulmonary Tuberculosis among People Living with Human Immunodeficiency Virus in Addis Ababa, Ethiopia: A Retrospective Cohort Study.

 Editor Comments:

1. Kindly thoroughly copyedit the manuscript for error using a native speaker

Thank you for the valuable comment. The revised manuscript is reviewed by someone who has better English language skills.

2. Why is reference 6 in CAPs?

 Thank you and we corrected it in the revised manuscript.

3. Need use to be consistent referencing style. Some references list only 3 while others list 6. Eg Ref 7 and 9

 Thank you for the comment, now corrected in the revised manuscript.

4. Ref 15. Remove typo 15

 Thank you and we removed it in the revised manuscript.

 Reviewers' comments

Thank you the reviewers for your valuable suggestions, constructive comments and the clarifications you want. 

Reviewer #1

 I read with interest this article. My main comment is that the language needs to be revised.

Thank you. The revised manuscript is reviewed by someone with better English language skill.

Other comments

1. The authors should provide information on the number of lost to follow up and transferred.

Thank you for the valuable comment, we included in the revised manuscript in line 199-201. “Among the remaining 494 patients who did not develop EPTB, 420 were on follow-up, 48 were lost to follow-up, 21 were transferred out and 5 died”.

2. The authors should also provide information on the modality of diagnosis, in how many was clinical and how many supported by lab/other diagnostic. This also deserves to be included in the discussion since extrapulmonary diagnosis is not easy especially in certain settings

Thank you for the comment, it is described in the revised manuscript in line 2004-2009.

“Bacteriological, histological and clinical methods were used for the diagnosis of EPTB. Accordingly, nine results were reported positive by bacteriological methods (Culture; 3, Xpert MTB/RIF assay; 6). Fifty-five results were confirmed by histological diagnosis and 16 test results were confirmed based on a combination of clinical and chest radiography. However, different combinations of the above confirmatory methods were used to rule out EPTB” We also included in the discussion part in line 268-270.

3. The authors should provide a limitation section in the discussion, including the lack of information of outcomes in the lost and transferred

Thank you for the comment we included in the revised manuscript in line 312-317. 

4. Conclusions sections. These are not conclusions, but a summary of the results. The authors should provide appropriate conclusion and recommendations. 

Thank you for the constructive comments. We corrected accordingly in the abstract section and the conclusion section.

Reviewer #2: 

Methods:

1. Please clarify the meaning of "disclosure status".

Thank you for the comment and we included it in the revised manuscript in line 141-142.

2. How was addiction to smoking and alcohol defined? Can the authors provide information on smoking status (never vs former vs current), pack-years smoked and quantitative estimate of alcohol consumption? Were these measures associated with EPTB?

Thank you for the comment it is operationalized in line 143. The bi-variable analysis was performed for all variables listed in this study, however only those with a P-Value <0.25 were subjected to multi-variable analysis. 

3. How was functional status defined/classified?

Thank you for the question we included it in the revised manuscript in line 144.

4. Were longitudinal CD4 and HIV RNA levels available during follow-up? Were these associated with the outcome?

Thank you for valuable comment. However, we were unable to get enough data to perform analysis because RNA/Viral load/ was started late and, thus no baseline viral load data available. Likewise, CD4 count was available at baseline, but not enough during the follow –up period and at the end of the follow-up period which is replaced by viral load count. In the revised manuscript we included it in the discussion and limitation part.

5. How many participants had incident PULMONARY TB disease and how was this handled in the analysis? Also, were prevalent pulmonary TB cases excluded?

Thank you for the clarification you want. During the follow-up period, there were 72 patients had incident pulmonary TB without having extrapulmonary TB and were followed until developed extrapulmonary TB or became censored.

6. Please provide the statistical method used to calculate incidence rate and its 95% confidence interval.

Thank you for the comment and we included it in the revised manuscript in line 159-164. “The EPTB incidence density rate per 100PYs of follow-up in each category is determined by dividing the number of EPTB in each category and multiplying by 100. The 95%CI for the incidence rate is determined using the formula 95%CI= eln incidence rate-Zα/2*SE, eln incidence rate +Zα/2*SE”

7. How was "IPT completion" defined?

Thank you for the comment and we included it in the revised manuscript in line 148-149. 

8. Are data on TB infection status available?

Thank you for the question. Previous TB history was assessed and included in the analysis as one risk factor but in the multivariable analysis it was not statistically significant.

9. How was "ART interruption" and "ART adherence" defined?

. We included in the revised manuscript under the operational definition section in line 150-154. 

10. Is duration of ART prior to enrollment available? Was this associated with incident EPTB?

Thank you for the question, unfortunately, it is not available. 

11. Did participants who died or were lost to follow-up different from the rest? And was death due to EPTB assessed as an outcome? It may be helpful to consider death and loss-to-follow-up as competing-risks for incident EPTB and analyze these data accordingly.

Thank you for the question and suggestion. “During the follow-up period there were 420 on follow-up, 48 lost to follow-up, 21 transferred out and 5 deaths which are considered as censored. Thus, follow-up data collected from these patients were included in the analysis being pooled as censored. The data collected from lost to follow-up and lost were not different from the rest participants. Death outcome due to EPTB was not assessed in the current study; however, your suggestion is very important”. 

Results: 

1. How many EPTB cases were defined by clinical vs microbiological diagnosis?

Thank you for the question and we included it in the revised manuscript.

2. Please report median and inter-quartile range or mean with standard deviation for all continuous exposures.

Thank you for the comment. We presented in the revised manuscript.

3. Please provide % in Figure-1.

Thank you for the comment and we included it in the revised manuscript. 

4. Table-1 and table-2 may be more informative if stratified by the primary outcome.

Thank you for your suggestion, this is done in table 3.

5. Is "tobacco smoking" in table-1 ever-smoking? Please clarify

Thank you for the question. It is ever smoking and we included in the operational definition. 

6. Please include foot-notes to all tables and figures clarifying abbreviations.

Thank you for the comment and the corrected accordingly in the revised manuscript.

7. Please clarify what "co-infection"means in Table-2.

Thank you for the comment, included in the revised manuscript in the footnote.

8. Please provide 95% confidence intervals for all incidence rate estimates in the text and tables

Thank you for the comment and we included it in the revised manuscript.

9. Was alcohol included in uni- and multi-variable analysis?

It is included in the bi-variable analysis and since the P-value was< 0.25 we did not include in the multivariable analysis.

Discussion:

1. Can the authors comment on the heterogeneity of HIV populations across the several ART clinics in Ethiopia and comment on the generalizability of their study findings?

Thank you for the suggestion. Even though HIV populations are homogeneous across several ART clinics of Ethiopia, the current study setting is the most populous and congregated city in the country. Thus this may favor TB infection thus generalizing for all HIV populations across the country might be difficult.

2. It is unclear why IRIS would lead to incident EPTB. Can the authors clarify?

Thank you for the comment. We included it in the revised manuscript in line 260-261.

3. The conclusion that male sex and not receiving co-trimoxazole prophylaxis are predictors of incident EPTB is incorrect. This may be due to confounding as seen in the uni- and multi-variable regression. Can the authors comment?

Thank you for the comment and we removed it 

4. Can the authors speculate as to why anemia may be associated with incident EPTB?

Thank you for the comment. We speculated in the revised manuscript.

5. The authors conclude that "early start of HAART" and "early diagnosis of HIV" should be considered for preventing incident EPTB. This is not supported by the data presented. Can the authors comment on the CD4 cell count at ART initiation (as opposed to enrollment) and whether this as associated with EPTB?

Thank you for the comment. We revised the conclusion.

General:

1. There are several typos in the manuscript text and figures. Please review and correct.

Thank you for valuable suggestion. We corrected it in the revised manuscript.

2. Recognizing that English may not be the authors primary language, please consider having this manuscript proof-read for grammatical errors and syntax.

Thank you for the valuable comment. The revised manuscript is reviewed by someone who has better English language skills.

---

## [Editor Report · Decision Letter 1]

15 Apr 2020

Incidence and Predictors of Extrapulmonary Tuberculosis among People Living with Human Immunodeficiency Virus in Addis Ababa, Ethiopia: A Retrospective Cohort study

PONE-D-20-00998R1

Dear Mr Alemu

We are pleased to inform you that your manuscript has been judged scientifically suitable for publication and will be formally accepted for publication once it complies with all outstanding technical requirements.

With kind regards,

Professor Kwasi Torpey, MD PhD MPH

Academic Editor

PLOS ONE
---

## [Editor Report · Acceptance letter]

20 Apr 2020

PONE-D-20-00998R1 

Incidence and Predictors of Extrapulmonary Tuberculosis among People Living with *Human Immunodeficiency Virus* in Addis Ababa, Ethiopia: A Retrospective Cohort study 

Dear Dr. Alemu:

I am pleased to inform you that your manuscript has been deemed suitable for publication in PLOS ONE. Congratulations! Your manuscript is now with our production department. 

With kind regards,

on behalf of

Professor Kwasi Torpey 

Academic Editor

PLOS ONE